# The impact of perioperative positive fluid balance on postoperative acute kidney injury in patients undergoing open hepatectomy: A retrospective single center cohort study

**Natsuda Phothikun**[1]☯*, **Orapan Pantatong**[1]☯, **Maytinee Kulpanun**[1]☯, **Somchai Wongpunkamol**[1]☯, **Worakitti Lapisatepun**[2,5]☯, **Amarit Phothikun**[3,4,5]☯, **Warangkana Lapisatepun**[1,4]☯

1 Department of Anesthesiology, Faculty of Medicine, Chiang Mai University, Chiang Mai, Thailand, 2 Division of Hepato-biliary and Pancreas, Department of Surgery, Faculty of Medicine, Chiang Mai University, Chiang Mai, Thailand, 3 Clinical Surgical Research Center, Chiang Mai University, Chiang Mai, Thailand, 4 Cardiovascular Thoracic Unit, Department of Surgery, Faculty of Medicine, Chiang Mai University, Chiang Mai, Thailand, 5 Department of Bioinformatics and Clinical Epidemiology, Faculty of Medicine, Chiang Mai University, Chiang Mai, Thailand

☯ These authors contributed equally to this work.
* natsuda.dkptk@gmail.com

## Abstract

### Background

Low central venous pressure (CVP) or fluid restriction strategies are frequently employed during liver parenchymal resection to minimize intraoperative blood loss. However, both hypovolemia and excessive fluid administration can impair organ perfusion, increasing the risk of renal dysfunction and acute kidney injury (AKI). This study explores the relationship between perioperative fluid management strategies and renal outcomes in patients undergoing hepatectomy.

### Method

A retrospective single-center cohort study was conducted involving 691 patients who underwent an open hepatectomy. Patients were categorized by positive fluid balance: <1 Liter, 1-2 Liters, and >2 Liters. Propensity score was used for matching among the groups. The incidence of acute kidney injury (AKI) was compared. Multivariable logistic regression analyzed the correlation between fluid balance and AKI risk.

### Result

The overall incidence of AKI was 11.58%, with the highest occurrence in the group with a fluid balance greater than 2 Liters. This group demonstrated a significantly higher relative risk of developing AKI compared to those with positive fluid balances of <1 Liter and 1-2 Liters (adjusted RR 1.85, p = 0.042, 95% CI 1.02-3.38). An increase in fluid balance was associated with a higher incidence rate ratio for AKI (p = 0.016). Additionally, an operating

**Data availability statement:** All relevant data are within the paper and its Supporting Information files.

**Funding:** This work was supported by Faculty of Medicine, Chiang Mai University, Thailand, grant no109-2564. This research did not receive any specific grant from funding agencies in the commercial. The funders had no role in study design, data collection and analysis, decision to publish, or preparation of the manuscript.

**Competing interests:** The authors have declared that no competing interests exist.

time >5 hours, blood loss >1000 ml, and Child-Turcotte-Pugh class B and C were significantly associated with an increased risk of post-hepatectomy AKI.

## Conclusion

Maintaining a fluid balance of 1-2 liters during hepatectomy is crucial to reducing the risk of postoperative AKI, while balances above 2 liters significantly increase it. Prolonged operating times, high blood loss, and advanced liver disease also elevate AKI risk, emphasizing the need for careful fluid management.

## 1. Introduction

Liver surgery is associated with a high rate of complications and mortality, with mortality rates reported as high as 10% to 30% [1]. This high risk is primarily due to the liver's extensive vascularization, as it receives about 25% of the cardiac output, which significantly increases the risk of intraoperative hemorrhage. Additionally, patients undergoing liver surgery often have pre-existing liver conditions, such as cirrhosis, or have undergone prior chemotherapy for liver cancer [2]. These conditions contribute to increased perioperative morbidity and mortality rates. Consequently, these factors highlight the necessity for specialized surgical techniques and expertise that are specifically tailored to liver surgery, differentiating it from other surgical procedures [3].

The concept of low central venous pressure (CVP) anesthesia is well-established for reducing intraoperative blood loss and improving the surgical field's visibility during the liver parenchymal transection phase. This technique involves maintaining CVP below 5 mmHg through fluid restriction, diuresis, and the use of vasodilators [4,5]. During the "restrictive phase," vasopressors can be administered to maintain hemodynamic stability. Following the completion of the liver parenchymal resection, intravascular fluid supplementation should be employed in the "resuscitative phase". Fluid liberal strategies may be used during this phase to reduce the risk of inadequate organ and tissue perfusion [6].

However, perioperative fluid management in liver resection is complex and controversial. Determining the appropriate volume of fluid to administer is challenging. Inadequate fluid administration may lead to poor organ perfusion and tissue oxygenation, contributing to acute kidney injury (AKI) [7]. Conversely, excessive fluid administration can cause fluid overload, leading to increased central venous pressure, impaired renal function, and potentially resulting in AKI [8–10]. These challenges in fluid management significantly impact patient outcomes, with the incidence of AKI remaining high, affecting 10-20% of patients [11–13].

Patients undergoing liver resection face a considerable risk of postoperative acute kidney injury (AKI). Specifically, hepatectomy substantially impacts fluid and electrolyte balance. The type, quantity, and method of fluid administration are critical aspects of patient care. Fluid administration during surgery influences blood circulation to various organs, particularly the renal system, which is susceptible to complications[14]. Excessive fluid administration may elevate intraoperative bleeding, while insufficient fluid administration can precipitate dehydration, leading to inadequate blood supply to vital organs. Thus, appropriate and balanced fluid administration can help decrease mortality rates and mitigate complications during and after surgery.

Numerous strategies for fluid intervention during hepatectomy have been reported, but Yoshino's 2017 review of 113 related studies found no clear conclusions about the optimal

fluid quantity. [15]. While Wang et al. investigated restricted fluid administration in donor hepatectomy using the principle of low CVP, they found reduced intraoperative bleeding but no impact on kidney function [16]. Additionally, in 2018, Wang found no significant kidney function differences between groups receiving varying fluid volumes during and after surgery [17]. Weinberg et al. compare goal-directed fluid therapy (GDFT) in hepatectomy with conventional fluid therapy, finding lower volumes with GDFT but similar post-surgery acute kidney injury and complication rates [18]. Moreover Carrier et al. examined fluid administration in liver transplant patients and its correlation. They found no discrepancy in AKI incidence. However, patients with a positive fluid balance experienced more complications and higher mortality rates [19].

The optimal fluid therapy strategy during hepatectomy remains unclear, and the impact of varying fluid volumes on kidney function and the incidence of postoperative AKI has not been well-defined. This study aims to investigate the association between perioperative positive fluid balance and postoperative AKI in hepatectomy patients.

## 2. Method

### 2.1. Patients and setting

The research employed a retrospective design, gathering registry data from Maharaj Nakorn Chiang Mai Hospital, Faculty of Medicine, Chiang Mai University, Thailand. Approval for the study was obtained from the Research Ethics Committee No.4 of the Faculty of Medicine, Chiang Mai University (the ethics committee waived the requirement for informed consent) (STUDY CODE: ANE-2563-07459, Research ID: 7459). The data were collected from the medical records of patients aged 18-70 years with ASA physical status I-IV who underwent elective open right or left hepatectomy from January 2008 to December 2019. The data were accessed for research purposes from 18 August 2020 to 17 August 2021, and all data were fully anonymized. The clinical variables evaluated included patient demographics (gender, age, weight, and BMI), ASA classification, and comorbidities such as diabetes mellitus, hypertension, dyslipidemia, cardiovascular and non-cardiovascular diseases. Hepatic conditions were categorized as benign tumors, hepatocellular carcinoma (HCC), liver metastasis, cholangiocarcinoma (CCA), or other pathologies, alongside hepatitis status (Hepatitis B, Hepatitis C, or co-infection). Liver and kidney function were assessed using the Child-Turcotte-Pugh classification (CTP), MELD score, and CKD staging. Major hepatectomy typically requires total bilirubin levels to be below 3 mg/dL before proceeding. However, some patients with total bilirubin levels exceeding 3 mg/dL, even after preoperative drainage, may still undergo resection due to the urgency of cancer treatment. In cases of obstructive jaundice, the decision to proceed with surgery must carefully balance the potential for improved hepatic function following biliary decompression with the oncological imperative of resecting the liver tumor. These complex scenarios underscore the importance of thorough preoperative assessment, judicious patient selection, and meticulous perioperative management to optimize outcomes. Intraoperative data included the type of hepatectomy (minor or major) [20], anesthesia and operating times, estimated blood loss, significant intraoperative hypotension, and the use of norepinephrine and colloids. These variables were extracted from medical records and anesthetic charts, with data validated by independent reviewers for accuracy and relevance to perioperative outcomes. Patients who underwent living-donor hepatectomy for liver transplantation, diagnosed with chronic kidney disease stage 5 and end stage renal disease with renal replacement therapy, and those with unresolved anemia (hematocrit < 25%) prior to surgery were excluded from the study.

The study compared patient groups receiving different intraoperative fluid management strategies during surgery. Based on the "Anesthesiology" guidelines by the American Society of

Anesthesiologists, recommendations for perioperative fluid therapy in major surgery suggest a moderately liberal intravenous fluid regimen with an overall positive fluid balance of 1–2 liters at the end of surgery [21,22]. Subsequently, the study divided patients into three groups: 1) those with a positive fluid balance of 1-2 liters at the end of surgery (group 1-2 L), 2) those with a positive fluid balance of less than 1 liter at the end of surgery (group < 1L), and 3) those with a positive fluid balance of over 2 liters at the end of surgery (group> 2L). A total of 691 cases of hepatectomy were included in the study. Group 1-2 L had 261 cases (37.78%), group < 1L had 213 cases (30.82%), and group> 2L had 217 cases (31.40%).

## 2.2. Endpoint

The primary endpoint consisted of two aspects. Firstly, it aimed to compare the incidence of AKI among three groups and investigate the relationship between fluid balance and AKI risk. The diagnosis criteria for AKI were based on the KDIGO criteria of 2012 [23], which included criteria such as an increase in serum creatinine (sCr) by $\geq 0.3$ mg/dl ($\geq 26.5$ mmol/l) within 48 hours, sCr elevation to $\geq 1.5$ times baseline within the prior 7 days, or urine volume < 0.5 ml/kg/h for 6 hours. Secondly, the focus was on comparing renal function, represented by the estimated glomerular filtration rate (eGFR), among the three groups (eGFR is highlighted as the most practical and useful laboratory value for evaluating renal function [24]).

For the secondary endpoint, the aim was to investigate the risk factors associated with AKI in hepatectomy patients.

## 2.3. Statistical analysis

The program STATA version 16.1 was utilized for analysis. Categorical data were compared and analyzed using the Chi-square test, while continuous data underwent analysis of variance (ANOVA) with Bonferroni correction for comparisons among the three groups.

Propensity score matching for multiple arms (three groups) was employed to minimize differences in patients' baseline characteristics. The propensity score in this study represented the predicted probability of receiving intraoperative fluid balance in positive balance categories of < 1L, 1-2L, and> 2L. The score was calculated using a multinomial logistic regression model [25], incorporating age, gender (male), BMI, CKD staging, and Child-Turcotte-Pugh classification for matching.

Regarding the outcomes, the risk of acute kidney injury associated with different fluid balances was analyzed using binomial (risk) regression. The incident rate ratio of acute kidney injury was assessed using linear regression. Differences in post-operative eGFR were compared using repeated measure regression. Additionally, the risk factors for AKI in hepatectomy patients were analyzed using univariable and multivariable logistic regression models.

All statistical differences were deemed significant at $p < 0.05$.

## 3. Result

### 3.1. Patients' demographic and intra-operative data

After matching with the propensity scores, the total number of cases was 579, with an equal distribution of 193 cases in all three groups (Table 1). There were no differences in demographic variables among the patient groups. Regarding operative data, the group with volumes greater than 2L exhibited the highest values for anesthetic time, operating time, mean estimated blood loss, significant intra-operative hypotension occurrences, percentage of norepinephrine usage, and percentage of colloid usage. (primarily starch-based solutions and gelatin-based solutions).

**Table 1. Demographic data and Intra-operative data.**

| Variables | All data | | | | Propensity score matching | | | |
|---|---|---|---|---|---|---|---|---|
| | < 1 L Group (n = 213) | 1-2 L Group (n = 261) | > 2 L Group (n = 217) | p | < 1 L Group (n = 193) | 1-2 L Group (n = 193) | > 2 L Group (n = 193) | p |
| Male | 127 (59.6) | 141 (54.0) | 140 (64.5) | 0.066 | 123 (63.7) | 121 (62.7) | 123 (63.7) | 0.971 |
| Age (Year) | 55.9 ± 12.3 | 55.7 ± 11.8 | 55.2 ± 12.0 | 0.841 | 56.2 ± 12.1 | 56.1 ± 11.6 | 55.1 ± 11.9 | 0.812 |
| Weight (kg) | 57.2 ± 10.5 | 56.9 ± 11.8 | 56.6 ± 11.1 | 0.230 | 57.7 ± 10.4 | 58.0 ± 12.1 | 56.7 ± 11.1 | 0.104 |
| BMI (kg/m²) | 22.4 ± 3.3 | 22.5 ± 4.0 | 22.6 ± 4.1 | 0.004 | 22.5 ± 3.3 | 22.7 ± 4.3 | 22.6 ± 4.1 | 0.054 |
| ASA Class | | | | 0.816 | | | | 0.794 |
| I | 31 (14.6) | 38 (14.6) | 32 (14.7) | | 24 (12.4) | 24 (12.4) | 32 (16.6) | |
| II | 154 (72.3) | 187 (71.6) | 154 (71.0) | | 143 (74.1) | 147 (76.2) | 137 (71.0) | |
| III | 25 (11.7) | 26 (10.0) | 24 (11.1) | | 21 (10.9) | 24 (12.4) | 21 (10.9) | |
| IV | 4 (1.4) | 10 (3.8) | 7 (3.2) | | 1 (0.5) | 2 (1.0) | 3 (1.5) | |
| Underlying disease | | | | | | | | |
| DM type2 | 26 (12.2) | 24 (9.2) | 28 (12.9) | 0.389 | 26 (13.5) | 16 (8.3) | 24 (12.4) | 0.238 |
| Hypertension | 66 (31.0) | 57 (21.8) | 51 (23.5) | 0.058 | 60 (31.1) | 44 (22.8) | 45 (23.3) | 0.113 |
| Dyslipidemia | 22 (10.3) | 24 (9.2) | 16 (7.4) | 0.556 | 20 (10.4) | 18 (9.3) | 15 (7.8) | 0.674 |
| CVS | 5 (2.4) | 13 (4.9) | 11 (5.1) | 0.270 | 5 (2.6) | 9 (4.7) | 11 (5.7) | 0.310 |
| Non-CVS | 59 (27.7) | 78 (29.9) | 63 (29.0) | 0.872 | 139 (72.0) | 129 (66.8) | 49 (25.4) | 0.229 |
| Hepatic disease | | | | 0.973 | | | | 0.993 |
| -Benign Tumor | 23 (10.8) | 27 (10.3) | 23 (10.7) | | 18 (9.3) | 19 (9.8) | 22 (11.5) | |
| -HCC | 82 (38.5) | 111(42.5) | 89 (41.2) | | 75 (38.9) | 82 (42.5) | 74 (38.5) | |
| -Liver metastasis | 16 (7.5) | 19 (7.3) | 14 (6.5) | | 15 (7.8) | 14 (7.3) | 13 (6.8) | |
| -CCA | 68 (31.9) | 77 (29.5) | 72 (33.3) | | 65 (33.7) | 60 (31.1) | 65 (33.9) | |
| -Others | 24 (11.3) | 27 (10.3) | 18 (8.3) | | 20 (10.4) | 18 (9.3) | 18 (9.4) | |
| Hepatitis | | | | 0.173 | | | | 0.224 |
| -Hepatitis B | 40 (18.8) | 38 (14.6) | 20 (9.2) | | 39 (20.2) | 28 (14.5) | 20 (10.4) | |
| -Hepatitis C | 6 (2.8) | 7 (2.7) | 4 (1.8) | | 6 (3.1) | 6 (3.1) | 4 (2.1) | |
| -Hepatitis B&C | 3 (1.4) | 3 (1.2) | 3 (1.4) | | 3 (1.6) | 3 (1.6) | 3 (1.6) | |
| Child-Turcotte-Pugh Classification | | | | 0.007 | | | | 0.998 |
| -A | 192 (90.1) | 227 (87.0) | 174 (80.2) | | 172 (89.1) | 171 (88.6) | 173 (89.7) | |
| -B | 20 (9.4) | 25 (9.6) | 38 (17.5) | | 20 (10.4) | 21 (10.9) | 19 (9.8 | |
| -C | 1 (0.5) | 9 (3.5) | 5 (2.3) | | 1 (0.5) | 1 (0.5) | 1 (0.5) | |
| CKD stage | | | | 0.857 | | | | 0.964 |
| -Stage1 | 53 (24.9) | 68 (26.1) | 54 (24.9) | | 48 (24.9) | 49 (25.4) | 48 (24.9) | |
| -Stage2 | 101 (47.4) | 114 (43.7) | 95 (43.8) | | 94 (48.7) | 90 (46.6) | 86 (44.6) | |
| -Stage3 | 56 (26.3) | 71 (27.2) | 64 (29.5) | | 48 (24.9) | 49 (25.4) | 55 (28.5) | |
| -Stage4 | 3 (1.4) | 8 (3.1) | 4 (1.8) | | 3 (1.6) | 5 (2.6) | 4 (2.1) | |
| Creatinine | 0.90 ± 0.38 | 0.89 ± 0.26 | 0.91 ± 0.29 | 0.611 | 0.91 ± 0.4 | 0.91 ± 0.3 | 0.92 ± 0.3 | 0.886 |
| Hemoglobin | 12.65 ± 1.8 | 12.32 ± 1.8 | 12.23 ± 2.2 | 0.063 | 12.67 ± 1.8 | 12.47 ± 1.9 | 12.27 ± 2.3 | 0.153 |
| Albumin | 4.03 ± 2.55 | 3.80 ± 0.61 | 3.68 ± 0.65 | 0.044 | 4.04 ± 2.67 | 3.81 ± 0.61 | 3.73 ± 0.63 | <0.001 |
| Bilirubin | 1.48 ± 5.10 | 1.45 ± 2.87 | 1.95 ± 2.88 | 0.272 | 1.52 ± 5.23 | 1.20 ± 2.23 | 1.38 ± 1.65 | <0.001 |
| MELD score | 5.4 ± 1.8 | 5.8 ± 2.1 | 6.4 ± 2.3 | 0.010 | 5.4 ± 1.9 | 5.7 ± 1.9 | 6.1 ± 2.1 | 0.241 |
| Intra-operative data | | | | | | | | |
| Hepatectomy type | | | | 0.207 | | | | 0.205 |
| -Minor | 96 (45.1) | 103 (39.5) | 80 (36.9) | | 90 (46.6) | 79 (40.9) | 73 (37.8) | |
| -Major | 117 (54.9) | 158 (60.5) | 137 (63.1) | | 103 (53.4) | 114 (59.1) | 120 (62.2) | |
| Anesthesia time (minutes) | 312.9 ± 110.9 | 368.7 ± 117.4 | 444.4 ± 136.4 | 0.006 | 312.7 ± 112.5 | 365.0 ± 120.4 | 434.8 ± 134.5 | 0.044 |
| Operating time (minutes) | 226.6 ± 105.2 | 319.6 ± 113.7 | 395.8 ± 133.1 | 0.002 | 266.6 ± 106.6 | 313.9 ± 115.9 | 386.9 ± 130.5 | 0.018 |

*(Continued)*

**Table 1.** (Continued)

| Variables | All data | | | | Propensity score matching | | | |
|---|---|---|---|---|---|---|---|---|
| | < 1 L Group (n = 213) | 1-2 L Group (n = 261) | > 2 L Group (n = 217) | p | < 1 L Group (n = 193) | 1-2 L Group (n = 193) | > 2 L Group (n = 193) | p |
| EBL (ml) | 772.6 ± 753.4 | 740.1 ± 531.8 | 1295.3 ± 989.0 | <0.001 | 758.4 ± 746.2 | 755.1 ± 558.8 | 1280.3 ± 988.6 | <0.001 |
| Significant intra-op hypotension | 139 (65.3) | 162 (62.1) | 166 (76.5) | 0.002 | 125 (64.8) | 124 (64.3) | 150 (77.7) | 0.005 |
| Intra-op Norepinephrine | 16 (7.5) | 19 (7.3) | 40 (18.4) | <0.001 | 14 (7.3) | 16 (8.29) | 39 (20.2) | <0.001 |
| RBC transfusion | 37 (17.4) | 73 (28.0) | 129 (59.5) | <0.001 | 34 (17.6) | 52 (26.9) | 118 (61.1) | <0.001 |
| FFP transfusion | 24 (11.3) | 39 (14.9) | 103 (47.5) | <0.001 | 20 (10.4) | 30 (15.5) | 87 (45.1) | <0.001 |
| Platelets transfusion | 5 (2.4) | 4 (1.5) | 9 (4.2) | 0.218 | 4 (2.1) | 2 (1.0) | 9 (4.7) | 0.098 |
| Colloid usage | 114 (53.5) | 182 (69.7) | 201 (92.6) | <0.001 | 105 (54.4) | 133 (68.9) | 179 (92.8) | <0.001 |
| Post-operative data | | | | | | | | |
| Hospital stays (days) | 10.4 ± 8.5 | 10.7 ± 9.4 | 11.3 ± 9.5 | 0.228 | 10.3 ± 8.6 | 10.5 ± 7.4 | 11.1 ± 9.6 | 0.001 |
| Mortality 30 days | 4 (1.88) | 7 (2.68) | 4 (1.84) | 0.847 | 4 (2.07) | 3 (1.55) | 4 (2.07) | 1.000 |
| Overall Mortality | 7 (3.29) | 10 (3.83) | 6 (2.76) | 0.803 | 7 (3.63) | 5 (2.59) | 6 (3.11) | 0.954 |

X (X); number (percent %), X ± X; Mean ± Standard deviation, L; Liters, Kg; Kilogram, m; meter, ml; milliliter, BMI; Body Mass Index, ASA class; American society of anesthesiologist classification, DM; Diabetes mellitus, CVS; Cardiovascular System diseases, non-CVS; other underlying diseases that do not involve the CVS, HCC; Hepatocellular carcinoma, CCA; Cholangiocarcinoma, CKD; Chronic kidney disease, MELD; Model for End-stage liver disease, EBL; Estimated blood loss, intra-op; intra operative, RBC; Red blood cell, FFP; Fresh frozen plasma.

Statistically significant at $p < 0.05$

## 3.2. Acute kidney injury

The overall incidence of AKI was 11.58%, with the highest incidence in the > 2L group, followed by the < 1L group, and the lowest in the 1-2L group (Table 2). According to the KDIGO criteria (S1 Table), the > 2L group had the highest number of AKI cases in every stage. Stage 1 AKI occurred in 9.84% of the < 1L group, 7.25% of the 1-2L group, and 14.51% of the > 2L group. Stage 2 AKI was found in 0.52% of the 1-2L group and 2.47% of the > 2L group. Stage 3 AKI was present only in the > 2L group at a rate of 0.52%. Additionally, among patients with CKD stage 1, the incidence of AKI was higher compared to other CKD stages.

When comparing the three groups, the group with a positive balance > 2L showed a statistically significant risk for AKI. Specifically, the > 2L group had a relative risk of developing AKI 1.85 times higher than those with positive balances < 1L and 1-2L (Table 3). Additionally, for every 1,000-milliliter increase in fluid balance, there was a corresponding increase in the incidence rate ratio for AKI by approximately 15%, which was statistically significant. This suggests that increase in fluid balance was associated with a higher risk of developing AKI.

## 3.3. Kidney function

Post-operative mean eGFR did not differ among all three groups (mean eGFR: < 1L Group; 75.4, 1-2L Group; 74.6, and > 2L Group; 74.2 ml/min/1.73m²). The change between pre- and post-operative eGFR was then compared. In Table 4, the mean eGFR decreased from pre- to post-operative stages in groups < 1L, 1-2L, and > 2L by 5.27, 7.84, and 2.99 mL/min/1.73m², respectively. A statistically significant difference was observed when comparing the change between the 1-2L group and the > 2L group. This indicates that the mean eGFR decline during pre- and post-operative stages was higher in the 1-2L group compared to the > 2L group. However, since the eGFR changes are minor, their clinical significance might be limited. It was more relevant to observe the original post-operative eGFR value, which did not show significant differences among the three groups (p = 0.082) (Table 4, Fig 1, and S2Table).

**Table 2. The incidence of acute kidney injury compared between original patients' data and propensity score matched data.**

| | Non-Matched AKI | | | Matched AKI | |
|---|---|---|---|---|---|
| | (n = 691) | p | | (n = 579) | p |
| Positive Balance | | 0.013 | | | 0.014 |
| < 1 L | 20/213 (9.39) | | | 19/193 (9.84) | |
| 1-2 L | 23/261 (8.81) | | | 15/193 (7.77) | |
| > 2 L | 37/217 (17.05) | | | 33/193 (17.1) | |
| CKD, n/total (%) | 80/691 (11.58) | 0.097 | | 67/579 (11.57) | 0.548 |
| -Stage1 | 29/175 (16.57) | | | 21/145 (14.48) | |
| -Stage2 | 34/310 (10.97) | | | 31/270 (11.48) | |
| -Stage3 | 16/191 (8.38) | | | 14/152 (9.21) | |
| -Stage4 | 1/15 (6.67) | | | 1/12 (8.33) | |
| Child-Turcotte-Pugh classification | | 0.001 | | | 0.001 |
| -A | 58/593 (9.78) | | | 51/516 (9.88) | |
| -B | 17/83 (20.48) | | | 14/60 (23.33) | |
| -C | 5/15 (33.33) | | | 2/3 (66.67) | |
| Type of hepatectomy | | 0.146 | | | 0.117 |
| -Minor hepatectomy | 26/279 (9.32) | | | 23/242 (9.5) | |
| -Major hepatectomy | 54/412 (13.11) | | | 44/337 (13.6) | |

X/X (X); number of AKI/ total of patients (percent % of AKI),

AKI; Acute kidney injury, L; Liters, CKD; Chronic kidney disease

Statistically significant at $p < 0.05$

**Table 3. Risk ratio of acute kidney injury by different balance (Propensity score matched) groups and incidence rate ratio of acute kidney injury.**

| Risk Ratio of acute kidney injury | | | | | | |
|---|---|---|---|---|---|---|
| | RR | p | 95%CI | Adjusted RR | p | 95%CI |
| Acute Kidney injury | | | | | | |
| -Positive Balance < 1 L | 1.26 | 0.474 | 0.66, 2.41 | 1.24 | 0.506 | 0.65, 2.38 |
| -Positive Balance 1-2 L | reference | | | | | |
| -Positive Balance > 2 L | 2.20 | 0.007 | 1.24, 3.91 | 1.85 | 0.042 | 1.02, 3.38 |
| Incidence rate ratio of acute kidney injury | | | | | | |
| | IRR | p | 95%CI | Adjusted IRR | p | 95%CI |
| Fluid Balance increase 1 ml | 1.00019 | <0.001 | 1.00011, 1.00028 | 1.00014 | 0.016 | 1.00002, 1.00025 |
| Fluid Balance increase 1,000 ml | 1.21 | <0.001 | 1.11, 1.32 | 1.15 | 0.016 | 1.03, 1.29 |

-Significant exploratory univariable of AKI variables = EBL, and Hypotension were used for adjusted in multivariable analysis.

-Correlations were shown between; AKI-Balance (0.125, p 0.002), AKI-EBL (0.124, p 0.002), AKI-hypotension (0.056, p 0.175), EBL-Balance (0.330, p < 0.001), EBL-hypotension (0.179, p < 0.001), hypotension-balance (0.133, p 0.001).

RR; risk ratio, CI; confidence interval, IRR; incidence rate ratio, L; Liters, AKI; acute kidney injury, EBL; estimated blood loss.

Statistically significant at p < 0.05

## 3.4. Risk factors of post-hepatectomy AKI

Multivariable logistic regression analysis reveals that patients classified as Child-Turcotte-Pugh class A were significantly associated with a decreased risk of post-hepatectomy AKI. Conversely, Child-Turcotte-Pugh classes B and C were significantly associated with an

**Table 4. post-operative estimated glomerular filtration rate analyzed by repeated measure regression.**

|  | Means change between pre- & post-operative | p | 95%CI |
|---|---|---|---|
| eGFR |  |  |  |
| Positive Balance < 1 L | -5.27 | <0.001 | -8.13, -2.42 |
| Positive Balance 1-2 L | -7.84 | <0.001 | -10.8, -4.87 |
| Positive Balance > 2 L | -2.99 | 0.048 | -5.96, -0.02 |

-When using group 1-2L as the reference for comparison analysis, the p-values between group 1-2L and group < 1L were 0.212, between group 1-2L and group> 2L were 0.026, and between the three groups was 0.082.

L; Liters, eGFR; Estimated glomerular filtration rate (ml/min/1.73m$^2$), CI; confidence interval.

Statistically significant at $p < 0.05$

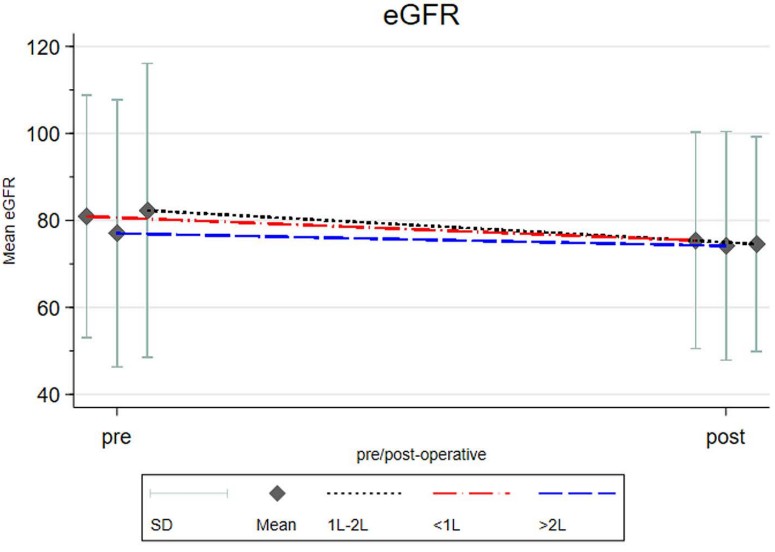

**Fig 1. Graph for the change of estimated glomerular filtration rate between groups.** L; Liters, eGFR; Estimated glomerular filtration rate (ml/min/1.73m$^2$), SD; standard deviation..

increased risk. Additionally, operative times exceeding 5 hours and estimated blood loss greater than 1000 ml were also significant risk factors for post-hepatectomy AKI (Table 5).

## 4. Discussion

In this retrospective, single-center cohort study of 691 patients undergoing open hepatectomy, approximately one in eleven patients developed AKI. After propensity score matching, patients with a positive fluid balance exceeding 2 L by the end of surgery had the highest incidence and relative risk of developing AKI compared to those with a positive fluid balance below 2 L. These findings underscore the importance of maintaining a fluid balance of 1–2 liters during hepatectomy to mitigate the risk of postoperative AKI. Prolonged operating times, significant blood loss, and advanced liver disease were also identified as key risk factors, emphasizing the need for careful patient selection and perioperative management.

Contrary to our findings, the RELIEF study concluded that in patients undergoing major abdominal surgery at increased risk of complications, a restrictive fluid regimen did not

**Table 5. Risk factors for acute kidney injury after hepatectomy.**

| Variables | Univariable analysis | | | Multivariable analysis | | |
|---|---|---|---|---|---|---|
| | OR | p | 95% CI | OR | p | 95% CI |
| Male | 0.71 | 0.223 | 0.41, 1.23 | 0.77 | 0.372 | 0.44, 1.35 |
| Age > 60 year | 0.78 | 0.373 | 0.45, 1.35 | 0.80 | 0.448 | 0.45, 1.41 |
| BMI < 18 kg/m² | 0.71 | 0.488 | 0.27, 1.85 | 0.65 | 0.375 | 0.24, 1.70 |
| BMI > 25 kg/m² | 0.66 | 0.235 | 0.34, 1.31 | 0.68 | 0.261 | 0.34, 1.34 |
| Albumin < 2.8 g/dL | 1.39 | 0.510 | 0.52. 3.74 | 1.00 | 1.000 | 0.35, 2.84 |
| Child-Turcotte-Pugh | | | | | | |
| A | 0.32 | <0.001 | 0.17, 0.61 | 0.34 | 0.001 | 0.17, 0.65 |
| B | 2.67 | 0.004 | 1.37, 5.16 | 2.55 | 0.007 | 1.29, 5.05 |
| C | 15.7 | 0.025 | 1.41-17.82 | 15.6 | 0.029 | 1.32,18.87 |
| Chronic kidney stage | | | | | | |
| Stage 1 | 1.42 | 0.207 | 0.82, 2.49 | 1.42 | 0.223 | 0.81, 2.53 |
| Stage 2 | 0.98 | 0.949 | 0.59, 1.64 | 1.00 | 0.990 | 0.60, 1.68 |
| Stage 3 | 0.72 | 0.291 | 0.38, 1.33 | 0.70 | 0.273 | 0.37, 1.32 |
| Stage 4 | 0.69 | 0.725 | 0.09, 5.43 | 0.69 | 0.729 | 0.09, 5.55 |
| Operation time > 5 hours | 2.23 | 0.003 | 1.33, 3.91 | 1.79 | 0.044 | 1.01, 3.19 |
| Anesthetic time > 5 hours | 1.88 | 0.045 | 1.01, 3.48 | 1.66 | 0.120 | 0.87, 3.15 |
| Significant Intra-op hypotension | 1.50 | 0.178 | 0.83, 2.71 | 0.95 | 0.912 | 0.45, 2.03 |
| EBL > 500 ml | 1.69 | 0.067 | 0.96, 2.95 | 1.12 | 0.739 | 0.57, 2.18 |
| EBL > 1000 ml | 2.25 | 0.002 | 1.34, 3.77 | 1.86 | 0.026 | 1.08, 3.23 |
| EBL > 2000 ml | 1.69 | 0.229 | 0.72, 3.99 | 0.44 | 0.225 | 0.12, 1.66 |
| Colloid use | 2.13 | 0.028 | 1.08, 4.18 | 1.47 | 0.293 | 0.71, 3.07 |
| Starch usage | 1.45 | 0.182 | 0.84, 2.53 | 1.04 | 0.893 | 0.57, 1.91 |
| Gelatin usage | 1.64 | 0.109 | 0.89, 3.03 | 1.09 | 0.814 | 0.54, 2.17 |
| RBC transfusion | 1.47 | 0.144 | 0.87, 2.46 | 0.86 | 0.643 | 0.45, 1.62 |
| FFP transfusion | 1.21 | 0.512 | 0.68, 2.16 | 0.56 | 0.129 | 0.26, 1.19 |
| Platelet transfusion | 1.18 | 0.829 | 0.26, 5.35 | 0.34 | 0.250 | 0.05, 2.15 |
| Intra-op norepinephrine | 2.45 | 0.006 | 1.29, 4.64 | 1.78 | 0.101 | 0.89, 3.58 |

OR; Odd ratio, CI; Confident interval, Kg; Kilogram, m; meter, ml; milliliter, g; gram, dL; deciliters, BMI; Body Mass Index, EBL; Estimated blood loss, intra-op; intra operative, RBC; Red blood cell, FFP; Fresh frozen plasma.

Statistically significant at p < 0.05.

improve disability-free survival rates compared to a liberal regimen but was associated with higher AKI incidence. However, the RELIEF trial excluded patients undergoing major liver resections, making its findings inapplicable to this subgroup. The AKI incidence observed in our study aligns with previous research on major liver resections, which reported rates ranging from 10% to 15% [11,13,26]. Moreover, our findings are consistent with studies demonstrating an increased risk of AKI in patients managed with liberal fluid strategies [27,28].

Shin et al.[29] reported that liberal fluid administration exceeding 2,700 mL significantly increased the risk of AKI (HR 1.29, p = 0.001, 95% CI 1.14–1.46). They attributed this to elevated central venous pressure, leading to renal subcapsular pressure, reduced renal blood flow, decreased glomerular filtration, and renal parenchymal edema. Similarly, Miller et al. proposed that hypervolemia from excessive fluid administration elevates intravascular hydro-static pressure, triggering the release of atrial natriuretic peptides, which can damage the renal endothelial glycocalyx [30]. This glycocalyx damage compromises renal function and systemic

vascular integrity [31–33]. Excessive fluid administration exacerbates hepatic congestion by increasing portal venous pressure, causing sinusoidal congestion and dilation of sinusoidal fenestrae. This results in protein and fluid exudation into the perisinusoidal space, impairing oxygen and nutrient diffusion to hepatocytes. Consequently, liver function deteriorates, and the risk of biliary anastomosis complications, such as bile leaks and hepatic insufficiency, increases [34,35].

In this study, Child-Turcotte-Pugh (CTP) class B and C were significant predictors of AKI, while CTP class A was associated with reduced AKI risk. Previous studies corroborate these findings, with Muciño-Bermejo et al. and Plailaharn et al. reporting similar associations between CTP classification and postoperative kidney function [36,37]. The mechanism underlying biliary obstruction and its nephrotoxic effects involves inflammatory, obstructive, and hemodynamic changes, contributing to biliary-induced nephropathy [16].

Low BMI (<18 kg/m²) demonstrated a trend toward increased AKI risk, though it was not statistically significant in univariable or multivariable analyses. Despite this, low BMI remains a critical factor due to its association with malnutrition and perioperative risks. Studies have identified underweight patients as more prone to AKI due to diminished physiological reserves and impaired immune responses [38,39]. Evaluating nutritional status with preoperative screening tools could provide valuable insights into the role of malnutrition in perioperative AKI. We also found that prolonged operative duration emerged as a risk factor for AKI, potentially due to anesthesia-induced hypotension and surgical complexity, which can impair renal perfusion. Additionally, estimated blood loss exceeding 1,000 mL significantly increased AKI risk, likely due to hypovolemia, hemodynamic instability, and compromised renal function. Although the use of starch-based and gelatin-based colloids was not significantly associated with AKI in this study, large-scale evidence indicates dose-dependent renal toxicity with starch-based solutions [40]. Differences in starch formulations, lower dosages, or patient selection may explain the absence of significant findings in our cohort.

This study highlights the critical role of fluid management in patients undergoing major hepatectomy. Maintaining a positive fluid balance of 1–2 L minimizes AKI risk without compromising hemodynamic stability, while excessive fluid administration significantly increases AKI risk. Risk stratification, including preoperative assessments of CTP classification and eGFR, is essential for identifying high-risk patients. Strategies to reduce blood loss and optimize hemodynamic parameters during surgery are also critical for renal protection.

Despite its strengths, including the use of propensity score matching to reduce patient heterogeneity and standardized AKI assessment criteria, this study has limitations. The retrospective design may introduce data reliability issues, with incomplete or invalid data accounting for less than 10% of cases. Additionally, the single-center database limits generalizability, and intraoperative variables, such as blood loss estimations, may vary among anesthesiologists.

## 5. Conclusion

In conclusion, this retrospective study of 691 hepatectomy patients indicates that maintaining a fluid balance of 1–2 liters at the end of surgery significantly reduces AKI risk. Liberal fluid strategies and excessive fluid administration were associated with increased renal impairment. Key risk factors for AKI included CTP class B/C cirrhosis, prolonged operative times, and significant blood loss.

## Supporting information

**S1 Data. The minimal data set used for analyzed.**
(XLSX)

**S1 Table. Acute kidney injury stage by different balance (Propensity score matched) groups.**
(DOCX)

**S2 Table. Estimated glomerular filtration rate between groups.**
(DOCX)

## Acknowledgement

We thank Chiang Mai University English language team (CELT) staff for language editing assistance.

## Author contributions

**Conceptualization:** Natsuda Phothikun, Somchai Wongpunkamol, Worakitti Lapisatepun, Warangkana Lapisatepun.

**Data curation:** Natsuda Phothikun, Orapan Pantatong, Maytinee Kulpanun, Amarit Phothikun, Warangkana Lapisatepun.

**Formal analysis:** Natsuda Phothikun, Amarit Phothikun, Warangkana Lapisatepun.

**Funding acquisition:** Natsuda Phothikun.

**Investigation:** Natsuda Phothikun, Somchai Wongpunkamol, Warangkana Lapisatepun.

**Methodology:** Natsuda Phothikun, Amarit Phothikun, Warangkana Lapisatepun.

**Project administration:** Natsuda Phothikun, Warangkana Lapisatepun.

**Resources:** Natsuda Phothikun, Orapan Pantatong, Maytinee Kulpanun, Somchai Wongpunkamol.

**Software:** Natsuda Phothikun.

**Supervision:** Natsuda Phothikun, Somchai Wongpunkamol, Worakitti Lapisatepun, Amarit Phothikun, Warangkana Lapisatepun.

**Validation:** Natsuda Phothikun, Orapan Pantatong, Somchai Wongpunkamol, Amarit Phothikun, Warangkana Lapisatepun.

**Visualization:** Natsuda Phothikun, Worakitti Lapisatepun, Amarit Phothikun, Warangkana Lapisatepun.

**Writing – original draft:** Natsuda Phothikun, Amarit Phothikun, Warangkana Lapisatepun.

**Writing – review & editing:** Natsuda Phothikun, Orapan Pantatong, Maytinee Kulpanun, Somchai Wongpunkamol, Worakitti Lapisatepun, Amarit Phothikun, Warangkana Lapisatepun.

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
