## [Decision Letter · Decision Letter 0]

26 Nov 2024

PONE-D-24-35759The Impact of Perioperative Positive Fluid Balance on Postoperative Acute Kidney Injury in Patients Undergoing Hepatectomy: A Single Center Retrospective Cohort StudyPLOS ONE

Dear Dr. Phothikun,

Thank you for submitting your manuscript to PLOS ONE. After careful consideration, we feel that it has merit but does not fully meet PLOS ONE’s publication criteria as it currently stands. Therefore, we invite you to submit a revised version of the manuscript that addresses the points raised during the review process.

We look forward to receiving your revised manuscript.

Kind regards,

Academic Editor

PLOS ONE

Journal Requirements:

2. Thank you for stating the following financial disclosure: [This work was supported by Faculty of Medicine, Chiang Mai University, Thailand, grant no109-2564. This research did not receive any specific grant from funding agencies in the commercial.]. Please state what role the funders took in the study. If the funders had no role, please state: "The funders had no role in study design, data collection and analysis, decision to publish, or preparation of the manuscript." If this statement is not correct you must amend it as needed. Please include this amended Role of Funder statement in your cover letter; we will change the online submission form on your behalf.

4. We notice that your supplementary [S1 and S2 Table] are included in the manuscript file. Please remove them and upload them with the file type 'Supporting Information'. Please ensure that each Supporting Information file has a legend listed in the manuscript after the references list.

Additional Editor Comments:

Reviewers have raised several concerns about your manuscript. As the Editor, I agree with the reviewers' comments. Without a satisfactory revision, I cannot determine if your manuscript aligns with the acceptance criteria for the PLOS ONE. Enclosed are the reviewers' comments and suggestions. Both reviewers and I share concerns regarding the Methods, Results, and Discussion sections in their current state. Please thoroughly address each of the raised issues in the attached comments, point by point. Make the necessary changes in the manuscript accordingly.

Reviewers' comments:

Reviewer's Responses to Questions

**Comments to the Author**

1. Is the manuscript technically sound, and do the data support the conclusions?

Reviewer #1: Yes

Reviewer #2: Partly

Reviewer #3: No

2. Has the statistical analysis been performed appropriately and rigorously? 

Reviewer #1: Yes

Reviewer #2: No

Reviewer #3: I Don't Know

3. Have the authors made all data underlying the findings in their manuscript fully available?

Reviewer #1: Yes

Reviewer #2: Yes

Reviewer #3: No

4. Is the manuscript presented in an intelligible fashion and written in standard English?

Reviewer #1: Yes

Reviewer #2: Yes

Reviewer #3: No

5. Review Comments to the Author

Reviewer #1: In particular, I found the study very well constructed and with findings that are in line with current literature.

Congratulations. The association of acute kidney injury with positive fluid balance needs to be publicized and publicized.

Reviewer #2: Dear Authors,

The manuscript titled "The Impact of Perioperative Positive Fluid Balance on Postoperative Acute Kidney Injury in Patients Undergoing Hepatectomy: A Single Center Retrospective Cohort Study" presents an interesting evaluation of the ideal positive fluid balance during hepatectomy to prevent acute kidney injury.

After reviewing the manuscript, I have raised several important observations:

- How many patients were excluded based on the following criteria: living-donor hepatectomy for liver transplantation, chronic kidney disease stage 5 and end-stage renal disease with renal replacement therapy, and unresolved anemia (hematocrit < 25%) prior to surgery?

- The clinical variables evaluated should be clearly outlined in the Methods section.

- In the Methods section, what variables were used for patient matching in the propensity score matching (PSM)?

- What criteria were used for selecting variables for the multivariable analysis? Was it based on p<0.05 in the univariable analysis?

- In the Discussion section, the authors mention the potential impact of incomplete or invalid data due to the retrospective nature of the study. Could you clarify which specific data were missing and how many patients were lost for each variable?

Reviewer #3: Thank you for the opportunity to review the manuscript by Phothikun et al., titled “The Impact of Perioperative Positive Fluid Balance on Postoperative Acute Kidney Injury in Patients Undergoing Hepatectomy: A Single-Center Retrospective Cohort Study.” I commend the authors for addressing a clinically significant issue in perioperative management of hepatectomy patients. This study is well-conceived, leveraging a large sample size to explore the relationship between fluid balance and postoperative acute kidney injury (AKI).

While the study provides valuable insights, there are several methodological and reporting issues that require attention to enhance the validity and clarity of the findings. I hope the following comments and suggestions will help refine the manuscript and improve its competitiveness for publication in PLOS ONE.

ABSTRACT

1. The Background section could be restructured for greater impact. Consider rewording as follows:

“Low central venous pressure (CVP) or fluid restriction strategies are frequently employed during liver parenchymal resection to minimize intraoperative blood loss. However, both hypovolemia and excessive fluid administration can impair organ perfusion, increasing the risk of renal dysfunction and acute kidney injury (AKI). This study explores the relationship between perioperative fluid management strategies and renal outcomes in patients undergoing hepatectomy.”

2. Add the term “single-center” between “retrospective” and “cohort.”

3. Insert a period at the end of the last sentence in the Methods section.

INTRODUCTION

4. Line 79: Replace “There were many different strategies for fluid therapy during hepatectomy” with “Numerous strategies for fluid intervention during hepatectomy have been reported.”

5. Line 80: This sentence is incomplete and needs correction.

6. Lines 81-82: Ensure consistent use of abbreviations after their first introduction. Replace “central venous pressure”with CVP throughout.

7. Line 88: Revise to “However, patients with a positive fluid balance experienced more complications and higher mortality rates [19].”

8. Lines 90-93: Consider the following rewording for clarity: “The optimal fluid therapy strategy during hepatectomy remains unclear, and the impact of varying fluid volumes on kidney function and the incidence of postoperative AKI has not been well-defined. This study aims to investigate the association between perioperative positive fluid balance and postoperative AKI in hepatectomy patients.”

PATIENT AND SETTINGS

9. Clarify why patients with ASA class IV were included in Table 1 despite the stated exclusion criteria (ASA I-III). Five such patients are noted in the database. Revise or comment.

10. Address the inclusion of patients with Child-Turcotte-Pugh (CTP) class C cirrhosis, despite the general contraindication for elective major liver resection in these cases. Provide context or justification using evidence-based references, such as Suman & Carey (2006). The authors conclude that patients with Child-Turcotte Pugh class C undergoing major liver resection and Child-Turcotte-Pugh scores were significantly associated with an increased risk of post-hepatectomy AKI. Patients with Child-Turcotte-Pugh (CTP) class C cirrhosis face significantly elevated perioperative mortality rates when undergoing major liver resection. Most centres would not undertake major liver resection surgery with patients with CTP class C. Studies indicate that these patients experience mortality rates exceeding 50% in the short term. For instance, a systematic review reported a short-term mortality rate of 52% (95% confidence interval: 33.5–70.0%) for CTP class C patients undergoing cardiac surgery. Similarly, another study found that patients with CTP class C cirrhosis had a postoperative mortality rate above 70% following abdominal surgery. Given these substantial risks, elective major liver resection is generally contraindicated in patients with Child-Turcotte-Pugh class C cirrhosis. (See Suman A, Carey WD. Assessing the risk of surgery in patients with liver disease. Cleve Clin J Med. 2006 Apr;73(4):398-404. doi: 10.3949/ccjm.73.4.398. PMID: 16610401.) Please comment?

11. The database shows discrepancies: 11 patients with CTP class C cirrhosis are categorized as ASA class II. This appears inconsistent. Please comment on data accuracy and integrity.

12. Define the criteria used to classify resections as major or minor.

13. Provide a clear definition for “significant intraoperative hypotension.”

14. Explicitly outline the method for calculating “estimated blood loss.”

15. Define the term “Non-CVS” as listed in Table 1.

16. Discuss why the Charlson Comorbidity Index (CCI) was not used for risk stratification. As a more objective measure than the ASA score, the CCI may influence outcomes. Could its omission bias findings?

17. Provide baseline data for creatinine, hemoglobin, albumin, and bilirubin values.

18. Report the proportion of patients in each group receiving red blood cell transfusions, fresh frozen plasma (FFP), cryoprecipitate, or platelets.

19. Include length of hospital stay, mortality rates, and 30-day readmission rates for each group.

20. Discuss whether fluid balance correlated with adverse events, such as bile leaks, hepatic insufficiency, respiratory complications, bleeding, return to the operating room, sepsis, or wound infections.

21. Discuss what the unit's fluid management stategy is for patients undergoing hepatectomy. Is "low CVP" used as standard? Do patients receive advanced hemodynamic monitoring? If so, what protocol is applied?

RESULTS

21. Confirm whether there were any missing data in this retrospective study. If so:

o Identify the type of missingness (MCAR, MAR, MNAR).

o Explain how missing data were handled.

o Detail propensity-matching strategies, such as imputation or weighting.

22. Clarify the surgical approach (open, laparoscopic, laparoscopic-assisted, or robotic) and discuss potential confounding effects on AKI incidence.

23. Discuss the relationship between low BMI (<18 kg/m²) and AKI development, referencing relevant studies (e.g., DOI: 10.1097/ALN.0000000000005215). In the database, 91 patients had an extremely low body mass index of less than 18 kg/m2 and 35 patients had a BMI < 17 kg/m2, which typically indicates severe malnutrition and is associated with significantly increased perioperative risks, including impaired wound healing, infections, and AKI. Numerous studies report an extremely low BMI is indeed associated with a higher risk of developing AKI after surgery (see https://doi.org/10.1097/ALN.0000000000005215 and DOI: 10.1053/j.jrn.2023.01.005. Could the author comment on whether a low BMI is independently associated with AKI?

24. A detailed descriptions of the type/s of colloids used needs to be reported. Certain colloids are strongly associated with the development of postop operative AKI. Some large studies with > 44,000 surgical patients show a dose-dependent renal toxicity with starch with 21% greater risk of developing AKI vs. crystalloid. See DOI: 10.1097/ALN.0000000000000375

DISCUSSION

25. There are numerous typographical errors and errors with syntax that need to be corrected. A professional editing service would substantially enhance this manuscript and improve its competitiveness for publication in PLOS ONE. Consider services such as https://www.medicaljournaleditors.com. This is an excellent service that provides comprehensive editing services. Alternative services include Editage (https://www.editage.com)

26. Revise or delete the first paragraph of the discussion, as it does not align with the study findings.

27. Structure the discussion into the following sections for coherence:

o Key findings (1 paragraph)

o Relationship to existing literature (2–3 paragraphs)

o Clinical implications (2 paragraphs)

o Strengths and limitations (2 paragraphs)

o Conclusions (1 paragraph)

28. Discuss potential mechanisms linking excessive fluid administration (>2L) with increased AKI risk.

CONCLUSION

29. Revise the conclusion for clarity. Consider the following “This retrospective study of 691 hepatectomy patients suggests that maintaining a fluid balance of 1–2 liters at surgery’s end minimizes AKI risk. Fluid excess or liberal fluid strategies were associated with renal impairment and AKI. Additional risk factors for AKI included CTP class B/C cirrhosis, prolonged operative duration, and significant blood loss.”

6. PLOS authors have the option to publish the peer review history of their article (what does this mean? ). If published, this will include your full peer review and any attached files.

**Do you want your identity to be public for this peer review?** For information about this choice, including consent withdrawal, please see our Privacy Policy .

Reviewer #1: **Yes: ** BRENNO CARDOSO GOMES

Reviewer #2: **Yes: ** GILTON MARQUES FONSECA

Reviewer #3: No

---

## [Author Response · Author response to Decision Letter 0]

3 Jan 2025

Comment from reviewer 1

In particular, I found the study very well constructed and with findings that are in line with current literature. Congratulations. The association of acute kidney injury with positive fluid balance needs to be publicized and publicized.

Response

Thank you for your positive feedback and recognition of our study’s alignment with current literature. We agree that the association between acute kidney injury and positive fluid balance is a crucial topic deserving further attention, and we appreciate the opportunity to contribute to this important discussion. 

Comment from reviewer 2

The manuscript titled "The Impact of Perioperative Positive Fluid Balance on Postoperative Acute Kidney Injury in Patients Undergoing Hepatectomy: A Single Center Retrospective Cohort Study" presents an interesting evaluation of the ideal positive fluid balance during hepatectomy to prevent acute kidney injury.

After reviewing the manuscript, I have raised several important observations:

Reviewer #2:

Comment 2.1: How many patients were excluded based on the following criteria: living-donor hepatectomy for liver transplantation, chronic kidney disease stage 5 and end-stage renal disease with renal replacement therapy, and unresolved anemia (hematocrit < 25%) prior to surgery?

Response:

The exclusion criteria were applied during the initial patient selection. As a result, no patients meeting these criteria were included in the study. While we do not have exact numbers, approximately 30 cases were excluded due to end-stage renal disease or chronic kidney disease stage 5, and around 20 due to living-donor hepatectomy for liver transplantation. The number of cases excluded for unresolved anemia is unknown.

Comment 2.2: The clinical variables evaluated should be clearly outlined in the Methods section.

Response: Thank you for pointing this out.

The clinical variables evaluated included patient demographics (gender, age, weight, and BMI), ASA classification, and comorbidities such as diabetes mellitus, hypertension, dyslipidemia, cardiovascular and non-cardiovascular diseases. Hepatic conditions were categorized as benign tumors, hepatocellular carcinoma (HCC), liver metastasis, cholangiocarcinoma (CCA), or other pathologies, alongside hepatitis status (Hepatitis B, Hepatitis C, or co-infection). Liver and kidney function were assessed using the Child-Turcotte-Pugh classification, MELD score, and CKD staging. Intraoperative data included the type of hepatectomy (minor or major), anesthesia and operating times, estimated blood loss, significant intraoperative hypotension, and the use of norepinephrine and colloids. These variables were extracted from medical records and anesthetic charts, with data validated by independent reviewers for accuracy and relevance to perioperative outcomes.

Comment 2.3: In the Methods section, what variables were used for patient matching in the propensity score matching (PSM)?

Response: Thank you for pointing this out.

The variables age, gender(male), BMI, CKD staging, and Child-Turcotte-Pugh classification were used for matching.

Comment 2.4 What criteria were used for selecting variables for the multivariable analysis? Was it based on p<0.05 in the univariable analysis?

Response: Thank you for pointing this out.

The variables included in the multivariable analysis were primarily selected based on statistical significance in the univariable analysis (p < 0.05). This criterion was used to identify variables with a potential association with the outcome. However, clinical relevance and prior evidence supporting their association with the outcome were also considered to ensure a robust and meaningful multivariable model. For example, variables such as Child-Turcotte-Pugh classification and operation time >5 hours were included due to their strong univariable associations (p < 0.05) and established clinical significance.

This table shows the full multivariable analysis with all variables included.

Variables Univariable analysis Multivariable analysis

OR p 95% CI OR p 95% CI

Male 0.71 0.223 0.41, 1.23 0.77 0.372 0.44, 1.35

Age >60 year 0.78 0.373 0.45, 1.35 0.80 0.448 0.45, 1.41

BMI <18 kg/m2 0.71 0.488 0.27, 1.85 0.65 0.375 0.24, 1.70

BMI >25 kg/m2 0.66 0.235 0.34, 1.31 0.68 0.261 0.34, 1.34

Albumin < 2.8 g/dl

1.39 0.510 0.52. 3.74 1.00 1.000 0.35, 2.84

Child-Turcotte-Pugh

A 0.32 <0.001 0.17, 0.61 0.34 0.001 0.17, 0.65

B 2.67 0.004 1.37, 5.16 2.55 0.007 1.29, 5.05

C 15.7 0.025 1.41-17.82 15.6 0.029 1.32,18.87

Chronic kidney stage

Stage 1 1.42 0.207 0.82, 2.49 1.42 0.223 0.81, 2.53

Stage 2 0.98 0.949 0.59, 1.64 1.00 0.990 0.60, 1.68

Stage 3 0.72 0.291 0.38, 1.33 0.70 0.273 0.37, 1.32

Stage 4 0.69 0.725 0.09, 5.43 0.69 0.729 0.09, 5.55

Operation time > 5 hours 2.23 0.003 1.33, 3.91 1.79 0.044 1.01, 3.19

Anesthetic time > 5 hours 1.88 0.045 1.01, 3.48 1.66 0.120 0.87, 3.15

Significant Intra-op hypotension 1.50 0.178 0.83, 2.71 0.95 0.912 0.45, 2.03

EBL > 500 ml 1.69 0.067 0.96, 2.95 1.12 0.739 0.57, 2.18

EBL > 1000 ml 2.25 0.002 1.34, 3.77 1.86 0.026 1.08, 3.23

EBL > 2000 ml 1.69 0.229 0.72, 3.99 0.44 0.225 0.12, 1.66

Colloid use 2.13 0.028 1.08, 4.18 1.47 0.293 0.71, 3.07

Starch usage 1.45 0.182 0.84, 2.53 1.04 0.893 0.57, 1.91

Gelatin usage 1.64 0.109 0.89, 3.03 1.09 0.814 0.54, 2.17

RBC transfusion 1.47 0.144 0.87, 2.46 0.86 0.643 0.45, 1.62

FFP transfusion 1.21 0.512 0.68, 2.16 0.56 0.129 0.26, 1.19

Platelet transfusion 1.18 0.829 0.26, 5.35 0.34 0.250 0.05, 2.15

Intra-op norepinephrine 2.45 0.006 1.29, 4.64 1.78 0.101 0.89, 3.58

OR; Odd ratio, CI; Confident interval, Kg; Kilogram, m; meter, ml; milliliter, BMI; Body Mass Index, EBL; Estimated blood loss, intra-op; intra operative, RBC; Red blood cell, FFP; Fresh frozen plasma.

Statistically significant at p <0.05.

Comment 2.5: In the Discussion section, the authors mention the potential impact of incomplete or invalid data due to the retrospective nature of the study. Could you clarify which specific data were missing and how many patients were lost for each variable?

Response: Thank you for pointing this out.

While this retrospective study examined a considerable number of patients and adhered to gold standard criteria for AKI assessment, including the use of propensity scores to minimize patient heterogeneity, it also had some limitations. Firstly, due to its retrospective nature, certain key data points were incomplete or missing, particularly those related to the primary outcome. For example, cases with uncertain records of AKI diagnosis or incomplete documentation of postoperative creatinine levels accounted for 13.84% of the dataset (n = 111 patients) (starting with total 802 case). These cases were excluded from the final analysis, which may have introduced selection bias and impacted the reliability of the findings. Secondly, using a database from a single center might limit how applicable the results are to broader populations. Lastly, certain data points, such as estimations of intraoperative bleeding, were reliant on individual practices and could have varied among attending anesthesiologists, potentially affecting the study's outcomes.

A complete case analysis (non-imputation) approach was utilized. Imputation was deemed unnecessary, as thorough data exploration revealed that essential data had a missing rate of less than 10%. Specifically, for diagnosis variables, the missing rate was exceptionally low at 0.17%, while all other variables had no missing data after PS matching.

Variable Missing Total Percent Missing Variable Missing Total Percent Missing

Gender 0 579 0 Intake 0 579 0

Age 0 579 0 Output 0 579 0

Wtkg 0 579 0 balance 0 579 0

Htcm 0 579 0 hypotension 0 579 0

BMI 0 579 0 Ephedrine 0 579 0

ASA 0 579 0 NE 0 579 0

Diagnosis 1 579 0.17 Epinephrine 0 579 0

Hepatitis 0 579 0 Dopamine 0 579 0

ChildPH 0 579 0 Dobutamine 0 579 0

+UD 0 579 0 Reverse 0 579 0

DM 0 579 0 Extubation 0 579 0

HTN 0 579 0 hospstayd 0 579 0

DLP 0 579 0 Intubation~d 0 579 0

UDCardio 0 579 0 Postophosp~d 0 579 0

preHb 0 579 0 Reop 0 579 0

preHct 0 579 0 Deathinadm 0 579 0

preWBC 0 579 0 momortality 0 579 0

prePlt 0 579 0 Stroke 0 579 0

preBUN 0 579 0 delirium 0 579 0

preCr0 0 579 0 Atelect 0 579 0

AKI 0 579 0 Pneumonia 0 579 0

AKIstage 0 579 0 Pleuraleff 0 579 0

Alb 0 579 0 PE 0 579 0

ALP 0 579 0 Hypotension 0 579 0

AST 0 579 0 Ileus 0 579 0

ALT 0 579 0 UTI 0 579 0

DB 0 579 0 SSI 0 579 0

TB 0 579 0 DVT 0 579 0

Anestimemin 0 579 0 ARDS 0 579 0

Optimemin 0 579 0 Liverfailure 0 579 0

crystalloid 0 579 0 CardiacArrest 0 579 0

Colloid 0 579 0 stageCKD 0 579 0

Stratch 0 579 0 CKD 0 579 0

Gelatin 0 579 0 HD 0 579 0

Albumin 0 579 0 POsepsis 0 579 0

PRCtrans 0 579 0 Pobleed 0 579 0

FFPtrans 0 579 0 posbalance 0 579 0

plttrans 0 579 0 eGFR0 0 579 0

EBL 0 579 0 eGFR1 0 579 0

UOPml 0 579 0 

Comment from reviewer 3

Thank you for the opportunity to review the manuscript by Phothikun et al., titled “The Impact of Perioperative Positive Fluid Balance on Postoperative Acute Kidney Injury in Patients Undergoing Hepatectomy: A Single-Center Retrospective Cohort Study.” I commend the authors for addressing a clinically significant issue in perioperative management of hepatectomy patients. This study is well-conceived, leveraging a large sample size to explore the relationship between fluid balance and postoperative acute kidney injury (AKI).

While the study provides valuable insights, there are several methodological and reporting issues that require attention to enhance the validity and clarity of the findings. I hope the following comments and suggestions will help refine the manuscript and improve its competitiveness for publication in PLOS ONE.

Reviewer #3

Comment 3.1: The Background section could be restructured for greater impact. Consider rewording as follows:

“Low central venous pressure (CVP) or fluid restriction strategies are frequently employed during liver parenchymal resection to minimize intraoperative blood loss. However, both hypovolemia and excessive fluid administration can impair organ perfusion, increasing the risk of renal dysfunction and acute kidney injury (AKI). This study explores the relationship between perioperative fluid management strategies and renal outcomes in patients undergoing hepatectomy.”

Response: Thank you for pointing this out. We revised and updated the manuscript according to your recommendations.

Comment 3.2: Add the term “single-center” between “retrospective” and “cohort.”

Response: Thank you for pointing this out. We revised and updated the manuscript according to your recommendations.

Comment 3.3: Insert a period at the end of the last sentence in the Methods section.

Response: Thank you for pointing this out. We revised and updated the manuscript according to your recommendations.

Comment 3.4: Line 79: Replace “There were many different strategies for fluid therapy during hepatectomy” with “Numerous strategies for fluid intervention during hepatectomy have been reported.”

Response: Thank you for pointing this out. We revised and updated the manuscript according to your recommendations.

Comment 3.5: Line 80: This sentence is incomplete and needs correction.

Response: Thank you for pointing this out. We revised and updated the manuscript according to your recommendations.

Comment 3.6: Lines 81-82: Ensure consistent use of abbreviations after their first introduction. Replace “central venous pressure” with CVP throughout.

Response: Thank you for pointing this out. We revised and updated the manuscript according to your recommendations.

Comment 3.7: Line 88: Revise to “However, patients with a positive fluid balance experienced more complications and higher mortality rates [19].”

Response: Thank you for pointing this out. We revised and updated the manuscript according to your recommendations.

Comment 3.8: Lines 90-93: Consider the following rewording for clarity: “The optimal fluid therapy strategy during hepatectomy remains unclear, and the impact of varying fluid volumes on kidney function and the incidence of postoperative AKI has not been well-defined. This study aims to investigate the association between perioperative positive fluid balance and postoperative AKI in hepatectomy patients.”

Response: Thank you for pointing this out. We revised and updated the manuscript according to your recommendations.

Comment 3.9: Clarify why patients with ASA class IV were included in Table 1 despite the stated exclusion criteria (ASA I-III). Five such patients are noted in the database. Revise or comment.

Response: Thank you for pointing this out.

We initially planned to include only patients with ASA classes I to III in the study. However, after analyzing the results, we found no significant differences between the study groups when ASA class IV patients were included. Therefore, these patients were ultimately included in the study, although the inclusion criteria were not updated to reflect this adjustment. We revised and updated the manuscript according to your recommendations.

Comment 3.10: Address the inclusion of patients with Child-Turcotte-Pugh (CTP) class C cirrhosis, despite the general contraindication for elective major liver resection in these cases. Provide context or justification using evidence-based references, such as Suman & Carey (2006). The authors conclude that patients with Child-Turcotte Pugh class C undergoing major liver resection and Child-Turcotte-Pugh scores were significantly associated with an increased risk of post-hepatectomy AKI. Patients with Child-Turcotte-Pugh (CTP) class C cirrhosis face significantly elevated perioperative mortality rates when undergoing major liver resection. Most centres would not undertake major liver resection surgery with patients with CTP class C. Studies indicate that these patients experience mortality rates exceeding 50% in the short term. For instance, a systematic review reported a short-term mortality rate of 52% (95% confidence interval: 33.5–70.0%) for CTP class C patients undergoing cardiac surgery. Similarly, another study found that patients with CTP class C cirrhosis had a postoperative mortality rate above 70% following abdominal surgery. Given these substantial risks, elective major liver resection is generally contraindicated in patients with Child-Turcotte-Pugh class C cirrhosis. (See Suman A, Carey WD. Assessing the risk of surgery in patients with liver disease. Cleve Clin J Med. 2006 Apr;73(4):398-404. doi: 10.3949/ccjm.73.4.398. PMID: 16610401.) Please comment?

Response: Thank you to the reviewer for raising this important concern, which may indeed cause confusion among readers.

In principle, the Child-Turcotte-Pugh (CTP) score was originally designed to evaluate chronic liver disease in patients without complicating factors such as obstructive jaundice. In our cohort, however, several patients particularly those with cholangiocarcinoma (CCA) or hepatocellular carcinoma (HCC) with bile duct tumor thrombus presented with preoperative obstructive jaundice. In these cases, where major hepatectomy with hilar resection or tumor thrombectomy was mostly required, elevated total bilirubin and decreased albumin levels may have artificially inflated the CTP score, inaccurately classifying certain patients as CTP class C. Therefore, their high CTP scores did not necessarily reflect the full extent of intrinsic liver dysfunction, but rather the impact of biliary obstruction.

Furthermore, the major hepatectomy typically requires total bilirubin levels to be below 3 mg/dL before proceeding. However, some patients with total bilirubin exceeding 3 mg/dL, even after preoperative drainage, still underwent resection due to the urgent for cancer treatment. We acknowledge that major liver resection in patients with a true CTP class C cirrhosis is associated with high perioperative risk and is generally contraindicated, as outlined by Suman and Carey (2006). Nonetheless, in those with obstructive jaundice, the decision to operate must also take into account the potential for improving hepatic function af

---

## [Decision Letter · Decision Letter 1]

26 Jan 2025

PONE-D-24-35759R1The Impact of Perioperative Positive Fluid Balance on Postoperative Acute Kidney Injury in Patients Undergoing Open Hepatectomy: A Retrospective Single Center Cohort StudyPLOS ONE

Dear Dr. Phothikun,

Thank you for submitting your manuscript to PLOS ONE. After careful consideration, we feel that it has merit but does not fully meet PLOS ONE’s publication criteria as it currently stands. Therefore, we invite you to submit a revised version of the manuscript that addresses the points raised during the review process.

We look forward to receiving your revised manuscript.

Kind regards,

Academic Editor

PLOS ONE

Additional Editor Comments:

I would like to thank the authors for their response and the extensive revisions made to the manuscript. While the manuscript has been significantly improved, several concerns remain. Please find below the comments from the reviewer.

Reviewers' comments:

Reviewer's Responses to Questions

**Comments to the Author**

1. If the authors have adequately addressed your comments raised in a previous round of review and you feel that this manuscript is now acceptable for publication, you may indicate that here to bypass the “Comments to the Author” section, enter your conflict of interest statement in the “Confidential to Editor” section, and submit your "Accept" recommendation.

Reviewer #2: (No Response)

Reviewer #3: (No Response)

2. Is the manuscript technically sound, and do the data support the conclusions?

Reviewer #2: Partly

Reviewer #3: Partly

3. Has the statistical analysis been performed appropriately and rigorously? 

Reviewer #2: No

Reviewer #3: Yes

4. Have the authors made all data underlying the findings in their manuscript fully available?

Reviewer #2: Yes

Reviewer #3: Yes

5. Is the manuscript presented in an intelligible fashion and written in standard English?

Reviewer #2: Yes

Reviewer #3: No

6. Review Comments to the Author

Reviewer #2: Dear Authors,

Thank you for your detailed responses to the reviewer's comments on your manuscript titled "The Impact of Perioperative Positive Fluid Balance on Postoperative Acute Kidney Injury in Patients Undergoing Hepatectomy: A Single Center Retrospective Cohort Study."

While we appreciate the comprehensive answers provided, we noticed that the corresponding modifications have not been incorporated into the manuscript text. To ensure the reviewer can easily assess the revisions, we kindly request that all changes based on the reviewer's feedback be added directly to the manuscript.

Additionally, please highlight these modifications within the manuscript. This will help streamline the review process and ensure clarity regarding the updates made in response to the reviewers' comments.

We look forward to receiving the revised manuscript with the highlighted changes at your earliest convenience.

Reviewer #3: Thank you once again for inviting me to review this resubmission. The authors are to be commended for their major review. The revised manuscript is much stronger, however there are still substantial minor changes that should be made. Of these, the discussion section requires substantial editing to enhance its clarity, flow, and structure. I have taken the liberty of providing a detailed edit of the discussion, maintaining the writing style of the authors, while improving readability and coherence. Additionally, I have corrected a few factual inaccuracies, particularly concerning the findings of the RELIEF study, to ensure precision and alignment with the evidence.

I hope these revisions prove helpful in strengthening the study and making the discussion section more impactful. Overall, this is an impressive and well-executed retrospective study with significant findings that are likely to influence the practice of anaesthesia for hepatectomy. I am confident it will be highly cited and contribute meaningfully to the literature.

Minor comments

1. Line 30, Abstract The sentence “….involving 691 patients underwent hepatectomy…” is incomplete. Please change to “A retrospective single-center cohort study was conducted involving 691 patients underwent who underwent an open hepatectomy.”

2. Lines 228-234: Discussion section. This entire paragraph should be deleted. It detracts significantly from the manuscript.

3. Line 237: Discussion section. The authors state that “For instance, the RELIEF trial compared a liberal fluid regimen to a zero-balance approach during surgery and found a significant increase in AKI rates with the liberal strategy.” This is incorrect! The RELIEF study showed that among patients undergoing major abdominal surgery at increased risk for complications, a restrictive fluid regimen was associated with a higher rate of acute kidney injury.

4. Lines 226-341. Discussion section. The entire discussion section requires substantial editing to enhance its clarity, flow, and structure. I have taken the liberty of providing a detailed edit to maintain your writing style while improving readability and coherence. Additionally, I have corrected a few factual inaccuracies, particularly concerning the findings of the RELIEF study, to ensure precision and alignment with the evidence. I hope these revisions prove helpful in strengthening the study and making the discussion section more impactful. Overall, this is an impressive and well-executed retrospective study with significant findings that are likely to influence the practice of anaesthesia for hepatectomy. I am confident it will be highly cited and contribute meaningfully to the literature.

5.Discussion section. Lines 275-279. The authors discuss that fluid overload adversely affects the respiratory system by raising intravascular hydrostatic pressure, leading to pulmonary edema, pleural effusions, and impaired oxygenation. They also discuss damage to the endothelial glycocalyx further heightens susceptibility to infections and sepsis by compromising immune responses (36). Please consider deleting these sentences. This did not investigate pulmonary complications or the immune system in any way. Please update all references accordingly.

5. Please consider revising the entire discussion to read as follows:

In this retrospective, single-center cohort study of 691 patients undergoing open hepatectomy, approximately one in eleven patients developed acute kidney injury (AKI). After propensity score matching, patients with a positive fluid balance exceeding 2 L by the end of surgery had the highest incidence and relative risk of developing AKI compared to those with a positive fluid balance below 2 L. These findings underscore the importance of maintaining a fluid balance of 1–2 liters during hepatectomy to mitigate the risk of postoperative AKI. Prolonged operating times, significant blood loss, and advanced liver disease were also identified as key risk factors, emphasizing the need for careful patient selection and perioperative management.

Contrary to our findings, the RELIEF study concluded that in patients undergoing major abdominal surgery at increased risk of complications, a restrictive fluid regimen did not improve disability-free survival rates compared to a liberal regimen but was associated with higher AKI incidence. However, the RELIEF trial excluded patients undergoing major liver resections, making its findings inapplicable to this subgroup. The AKI incidence observed in our study aligns with previous research on major liver resections, which reported rates ranging from 10% to 15% (11, 13, 26). Moreover, our findings are consistent with studies demonstrating an increased risk of AKI in patients managed with liberal fluid strategies (27, 28).

Shin et al. (29) reported that liberal fluid administration exceeding 2,700 mL significantly increased the risk of AKI (HR 1.29, p = 0.001, 95% CI 1.14–1.46). They attributed this to elevated central venous pressure, leading to renal subcapsular pressure, reduced renal blood flow, decreased glomerular filtration, and renal parenchymal edema. Similarly, Miller et al. proposed that hypervolemia from excessive fluid administration elevates intravascular hydrostatic pressure, triggering the release of atrial natriuretic peptides, which can damage the renal endothelial glycocalyx (30). This glycocalyx damage compromises renal function and systemic vascular integrity (31–33). Excessive fluid administration exacerbates hepatic congestion by increasing portal venous pressure, causing sinusoidal congestion and dilation of sinusoidal fenestrae. This results in protein and fluid exudation into the perisinusoidal space, impairing oxygen and nutrient diffusion to hepatocytes. Consequently, liver function deteriorates, and the risk of biliary anastomosis complications, such as bile leaks and hepatic insufficiency, increases (34, 35).

In this study, Child-Turcotte-Pugh (CTP) class B and C were significant predictors of AKI, while CTP class A was associated with reduced AKI risk. Previous studies corroborate these findings, with Muciño-Bermejo et al. and Plailaharn et al. reporting similar associations between CTP classification and postoperative kidney function (37, 38). The mechanism underlying biliary obstruction and its nephrotoxic effects involves inflammatory, obstructive, and hemodynamic changes, contributing to biliary-induced nephropathy (16).

Low BMI (<18 kg/m²) demonstrated a trend toward increased AKI risk, though it was not statistically significant in univariable or multivariable analyses. Despite this, low BMI remains a critical factor due to its association with malnutrition and perioperative risks. Studies have identified underweight patients as more prone to AKI due to diminished physiological reserves and impaired immune responses (39, 40). Evaluating nutritional status with preoperative screening tools could provide valuable insights into the role of malnutrition in perioperative AKI. We also found that prolonged operative duration emerged as a risk factor for AKI, potentially due to anesthesia-induced hypotension and surgical complexity, which can impair renal perfusion. Additionally, estimated blood loss exceeding 1,000 mL significantly increased AKI risk, likely due to hypovolemia, hemodynamic instability, and compromised renal function. Although the use of starch-based and gelatin-based colloids was not significantly associated with AKI in this study, large-scale evidence indicates dose-dependent renal toxicity with starch-based solutions (41). Differences in starch formulations, lower dosages, or patient selection may explain the absence of significant findings in our cohort.

This study highlights the critical role of fluid management in patients undergoing major hepatectomy. Maintaining a positive fluid balance of 1–2 L minimizes AKI risk without compromising hemodynamic stability, while excessive fluid administration significantly increases AKI risk. Risk stratification, including preoperative assessments of CTP classification and eGFR, is essential for identifying high-risk patients. Strategies to reduce blood loss and optimize hemodynamic parameters during surgery are also critical for renal protection.

Despite its strengths, including the use of propensity score matching to reduce patient heterogeneity and standardized AKI assessment criteria, this study has limitations. The retrospective design may introduce data reliability issues, with incomplete or invalid data accounting for less than 10% of cases. Additionally, the single-center database limits generalizability, and intraoperative variables, such as blood loss estimations, may vary among anesthesiologists.

In conclusion, this retrospective study of 691 hepatectomy patients indicates that maintaining a fluid balance of 1–2 liters at the end of surgery significantly reduces AKI risk. Liberal fluid strategies and excessive fluid administration were associated with increased renal impairment. Key risk factors for AKI included CTP class B/C cirrhosis, prolonged operative times, and significant blood loss.

7. PLOS authors have the option to publish the peer review history of their article (what does this mean? ). If published, this will include your full peer review and any attached files.

**Do you want your identity to be public for this peer review?** For information about this choice, including consent withdrawal, please see our Privacy Policy .

Reviewer #2: **Yes: ** Gilton Marques Fonseca

Reviewer #3: **Yes: ** Professor Laurence Weinberg

---

## [Author Response · Author response to Decision Letter 1]

4 Feb 2025

A rebuttal letter.

Dear, Academic Editor

Thank you for giving us the opportunity to submit a 2nd revised draft of our manuscript titled “The Impact of Perioperative Positive Fluid Balance on Postoperative Acute Kidney Injury in Patients Undergoing Open Hepatectomy: A Retrospective Single Center Cohort Study” to PLOS ONE. We appreciate the time and effort you and the reviewers have dedicated to providing valuable feedback on our manuscript. We are grateful to the reviewers for their insightful comments on our paper. We have been able to incorporate changes to reflect most of the suggestions provided by the reviewers.

Here is a point-by-point response to the reviewers’ comments and concerns.

Comments to the Author

Comment from reviewer 2

Thank you for your detailed responses to the reviewer's comments on your manuscript titled "The Impact of Perioperative Positive Fluid Balance on Postoperative Acute Kidney Injury in Patients Undergoing Hepatectomy: A Single Center Retrospective Cohort Study."

While we appreciate the comprehensive answers provided, we noticed that the corresponding modifications have not been incorporated into the manuscript text. To ensure the reviewer can easily assess the revisions, we kindly request that all changes based on the reviewer's feedback be added directly to the manuscript.

Additionally, please highlight these modifications within the manuscript. This will help streamline the review process and ensure clarity regarding the updates made in response to the reviewers' comments.

We look forward to receiving the revised manuscript with the highlighted changes at your earliest convenience.

Reviewer #2:

Response: Thank you for pointing this out.

All modifications have been incorporated into the manuscript text. To ensure the reviewer can easily assess the revisions, we have resubmitted the revised manuscript with tracked changes. Additionally, we have provided a version with highlighted modifications for clarity. Revisions based on Reviewer 2’s feedback are marked in green, while those based on Reviewer 3’s feedback are marked in yellow. 

Comment from reviewer 3

Thank you once again for inviting me to review this resubmission. The authors are to be commended for their major review. The revised manuscript is much stronger, however there are still substantial minor changes that should be made.

Of these, the discussion section requires substantial editing to enhance its clarity, flow, and structure. I have taken the liberty of providing a detailed edit of the discussion, maintaining the writing style of the authors, while improving readability and coherence. Additionally, I have corrected a few factual inaccuracies, particularly concerning the findings of the RELIEF study, to ensure precision and alignment with the evidence.

I hope these revisions prove helpful in strengthening the study and making the discussion section more impactful. Overall, this is an impressive and well-executed retrospective study with significant findings that are likely to influence the practice of anesthesia for hepatectomy. I am confident it will be highly cited and contribute meaningfully to the literature.

Response

Thank you for your thoughtful and constructive feedback. We sincerely appreciate your detailed edits to the discussion section, which have greatly improved its clarity, flow, and structure while preserving our writing style. Your corrections, particularly regarding the findings of the RELIEF study, have been invaluable in ensuring the accuracy and alignment of our discussion with the existing evidence.

We have carefully incorporated your suggested revisions and believe they have strengthened the manuscript significantly. Thank you once again for your time and expertise in reviewing our work. Your insightful comments and support are truly appreciated.

Reviewer #3

Comment 3.1: Line 30, Abstract The sentence “….involving 691 patients underwent hepatectomy…” is incomplete. Please change to “A retrospective single-center cohort study was conducted involving 691 patients underwent who underwent an open hepatectomy.”

Response: Thank you for pointing this out. We revised and updated the manuscript according to your recommendations.

Comment 3.2: Lines 228-234: Discussion section. This entire paragraph should be deleted. It detracts significantly from the manuscript.

Response: Thank you for pointing this out. We deleted and updated the manuscript according to your recommendations.

Comment 3.3: Line 237: Discussion section. The authors state that “For instance, the RELIEF trial compared a liberal fluid regimen to a zero-balance approach during surgery and found a significant increase in AKI rates with the liberal strategy.” This is incorrect! The RELIEF study showed that among patients undergoing major abdominal surgery at increased risk for complications, a restrictive fluid regimen was associated with a higher rate of acute kidney injury.

Response: Thank you for pointing this out. We revised and updated the manuscript according to your recommendations.

Comment 3.4: Lines 226-341. Discussion section. The entire discussion section requires substantial editing to enhance its clarity, flow, and structure. I have taken the liberty of providing a detailed edit to maintain your writing style while improving readability and coherence. Additionally, I have corrected a few factual inaccuracies, particularly concerning the findings of the RELIEF study, to ensure precision and alignment with the evidence. I hope these revisions prove helpful in strengthening the study and making the discussion section more impactful. Overall, this is an impressive and well-executed retrospective study with significant findings that are likely to influence the practice of anesthesia for hepatectomy. I am confident it will be highly cited and contribute meaningfully to the literature.

Response: Thank you for pointing this out. We revised and updated the manuscript according to your recommendations.

Comment 3.5: Discussion section. Lines 275-279. The authors discuss that fluid overload adversely affects the respiratory system by raising intravascular hydrostatic pressure, leading to pulmonary edema, pleural effusions, and impaired oxygenation. They also discuss damage to the endothelial glycocalyx further heightens susceptibility to infections and sepsis by compromising immune responses (36). Please consider deleting these sentences. This did not investigate pulmonary complications or the immune system in any way. Please update all references accordingly.

Response: Thank you for pointing this out. We revised and updated the manuscript according to your recommendations.

Comment 3.6: Please consider revising the entire discussion to read as follows: (Lines 226-341) …

Response: Thank you for pointing this out. We revised and updated the manuscript according to your recommendations.

We look forward to hearing from you in due time regarding our submission and to responding to any further questions and comments you may have.

Sincerely, yours

Natsuda Phothikun

Department of Anesthesiology, Faculty of Medicine,

Chiang Mai University, Chiang Mai, Thailand

110 Intravaroros Road, Sriphum, Mueng,

Chiang Mai 50200, Thailand

Email: natsuda.dkptk@gmail.com

Tel: +66849090104

---

## [Decision Letter · Decision Letter 2]

11 Feb 2025

The Impact of Perioperative Positive Fluid Balance on Postoperative Acute Kidney Injury in Patients Undergoing Open Hepatectomy: A Retrospective Single Center Cohort Study

PONE-D-24-35759R2

Dear Dr. Phothikun,

We’re pleased to inform you that your manuscript has been judged scientifically suitable for publication and will be formally accepted for publication once it meets all outstanding technical requirements.

Kind regards,

Young-Kug Kim, M.D., Ph.D.

Academic Editor

PLOS ONE

Additional Editor Comments (optional):

The authors responded satisfactorily to the questions of the reviewers. Based on this, we believe the paper contains important and new information.

Reviewers' comments:

Reviewer's Responses to Questions

**Comments to the Author**

1. If the authors have adequately addressed your comments raised in a previous round of review and you feel that this manuscript is now acceptable for publication, you may indicate that here to bypass the “Comments to the Author” section, enter your conflict of interest statement in the “Confidential to Editor” section, and submit your "Accept" recommendation.

Reviewer #2: All comments have been addressed

Reviewer #3: All comments have been addressed

2. Is the manuscript technically sound, and do the data support the conclusions?

Reviewer #2: Yes

Reviewer #3: Yes

3. Has the statistical analysis been performed appropriately and rigorously? 

Reviewer #2: Yes

Reviewer #3: Yes

4. Have the authors made all data underlying the findings in their manuscript fully available?

Reviewer #2: Yes

Reviewer #3: Yes

5. Is the manuscript presented in an intelligible fashion and written in standard English?

Reviewer #2: Yes

Reviewer #3: Yes

6. Review Comments to the Author

Reviewer #2: Dear Authors,

The manuscript titled "The Impact of Perioperative Positive Fluid Balance on Postoperative Acute Kidney Injury in Patients Undergoing Open Hepatectomy: A Retrospective Single-Center Cohort Study" has adequately addressed the raised concerns and is suitable for publication in its current form.

Reviewer #3: I am satisfied that all changes have been incorporated into this revision. Congratulations on an excellent manuscript.

7. PLOS authors have the option to publish the peer review history of their article (what does this mean? ). If published, this will include your full peer review and any attached files.

**Do you want your identity to be public for this peer review?** For information about this choice, including consent withdrawal, please see our Privacy Policy .

Reviewer #2: **Yes: ** Gilton Marques Fonseca

Reviewer #3: **Yes: ** Laurence Weinberg

---

## [Editor Report · Acceptance letter]

PONE-D-24-35759R2

PLOS ONE

Dear Dr. Phothikun,

I'm pleased to inform you that your manuscript has been deemed suitable for publication in PLOS ONE. Congratulations! Your manuscript is now being handed over to our production team.

Kind regards,

on behalf of

Prof. Young-Kug Kim

Academic Editor

PLOS ONE